# Evaluation of the effectiveness and practicality of erbium lasers for ceramic restoration removal: A retrospective clinical analysis

**Janina Golob Deeb[1], Kinga Grzech-Lesniak[1,2], Sompop Bencharit[3,4]***

1 Department of Periodontics, School of Dentistry, Virginia Commonwealth University, Richmond, VA, United States of America, 2 Department of Oral Surgery, Laser Laboratory, Wroclaw Medical University, Wroclaw, Poland, 3 Office of Oral Health Innovation, College of Dental Medicine, Medical University of South Carolina, Charleston, SC, United States of America, 4 Department of Reconstructive & Rehabilitation Sciences, College of Dental Medicine, Medical University of South Carolina, Charleston, SC, United States of America

* benchari@musc.edu

**Data Availability Statement:** All relevant data are within the manuscript.

**Funding:** The authors received no specific funding for this work.

## Abstract

### Background

The purpose of this study was to assess the effectiveness and practicality of erbium lasers in the removal of ceramic restorations and appliances from natural teeth and dental implant abutments in clinical practice.

### Methods

A retrospective analysis was conducted, involving 29 clinical cases with a total of 52 abutments requiring the removal of various ceramic restorations. The analysis evaluated the clinical procedures performed, including the type and material of the prosthetic, the type of cement used, laser setting parameters, retrieval time, and retrieval success.

### Results

Out of the 52 abutments, 50 were successfully retrieved without causing any damage (>95%) using either an Er,Cr:YSGG laser (N = 6) or an Er:YAG laser (N = 46). In one case, a crown was partially sectioned to prevent any negative impact of laser irradiation on the adhesive strength between the post and tooth, and in another case, a fracture occurred during debonding. The restorations consisted of 13 lithium disilicate and 39 zirconia units, including six veneers, 38 single crowns, and three fixed partial dentures (FPDs). The retrieval time varied depending on the restoration type, material thickness, cement type, retention form/fitting of the abutment and restoration, ranging from 2.25 ±0.61 minutes for veneers, 6.89 ±8.07 minutes for crowns, to 25 ±10 minutes per abutment for FPDs. Removal of a zirconia crown required more time, 7.12±8.91 minutes, compared to a lithium disilicate crown, 5.86 ±2.41 minutes. The debonding time was influenced by the laser settings as well as materials and types of prosthesis.

**Competing interests:** The authors have declared
that no competing interests exist.

## Conclusions

Erbium lasers present a safe and effective alternative to invasive methods for removing
ceramic restorations, without causing harm to the abutment or prosthesis. Laser-assisted
debonding allows for recementation of the restorations during the same appointment, mak-
ing it a conservative and viable option for ceramic crown retrieval in clinical settings.

## Introduction

In recent years, all-ceramic restorations, such as lithium disilicate and zirconia, have gained
prominence and become the preferred choice for restoring natural teeth and implants. This is
primarily due to their numerous benefits, including superior aesthetics, biocompatibility, and
reliable long-term durability [1,2] The clinical success of luting cements and bonded ceramic
restorations, combined with the strength of modern ceramic materials [3], ensures the long-
term survival of these restorations. However, when the time comes for the removal of such res-
torations, it poses a challenge for clinicians. Typically, the conventional removal of bonded or
cemented all-ceramic restorations involves the use of bur and handpiece techniques [4]. The
conventional technique for removing bonded or cemented all-ceramic restorations can vary in
cutting efficiency depending on the handpiece and bur used [5]. One of the challenges is that it
can be difficult to visually distinguish between the tooth-colored cement and the natural tooth
structure, leading to the common occurrence of abutment tooth damage or unnecessary
removal of tooth structure during conventional restorative removal [6,7]. Furthermore, even
after the crown is removed, clinicians often make efforts to remove most of the remaining
composite resin luting material, as any residual composite material can potentially interfere
with future dentin preparation for new bonded restorative material [8]. Unfortunately, the
conventional removal of cement-retained fixed prostheses, whether for natural teeth or dental
implant abutments, frequently results in irreversible damage to the restorations, rendering
them unusable.

Researchers have proposed the use of erbium lasers for the debonding of ceramic orthodon-
tic brackets and other all-ceramic restorations and prostheses [9,10]. Studies have shown that
all-ceramic CAD-CAM bonded lithium disilicate and zirconia crowns can be successfully
removed without noticeable structural damage to the restoration or prosthesis on natural teeth
[11], and implant abutments [12–14]. Similar positive results have been observed for prefabri-
cated zirconia crowns in pediatric patients [7,15], endodontic fiber-reinforced composite post
[16], and veneers [17–20]. In dentistry, two types of erbium lasers are commonly used:
Erbium, Chromium-doped Yttrium Scandium Gallium Garnet (Er,Cr:YSGG) and Erbium-
doped Yttrium Aluminum Garnet (Er:YAG) lasers. These lasers operate at wavelengths of
2780 nm and 2940 nm, respectively [21]. Laser-assisted debonding operates through the inter-
action of laser energy with residual monomers and water molecules present in the luting
cement material [13,14,22]. When the laser is applied to the ceramic restoration, the energy is
transmitted through the translucent ceramic material with minimal absorption. The laser
energy is selectively absorbed by specific chromophores, such as water and unpolymerized
monomers, resulting in their energization and vibration. This leads to the thermomechanical
ablation of the cement, causing its vaporization [13,14,22]. Both Er,Cr:YSGG and Er:YAG
lasers can be utilized for ceramic restoration debonding, although the Er:YAG laser generally
demonstrates greater effectiveness [14,15]. Due to the laser energy primarily being absorbed
by water and monomer molecules in the luting cement, as well as the presence of water coolant

during the debonding process, scanning electron microscopy (SEM) and energy dispersive spectroscopy (EDX) analyses have revealed no significant surface or chemical damage to natural teeth, implant abutments, or debonded ceramic restorations [7,11–14]. Importantly, the temperature changes resulting from laser applications do not appear to have any adverse effects on the dental pulp, implants, or surrounding bone [7,11–15].

Although laser-assisted debonding of all-ceramic restorations shows promise, the majority of studies in this field are conducted in vitro or utilize ex-vivo simulations or case reports [15]. Currently, there is a lack of standardized protocols or guidelines for the clinical application of this technique, and different operative settings have been proposed. Several factors, such as the chemical composition of the prosthesis and cement materials, thickness of the cement layer, opacity and thickness of the ceramic restoration, as well as laser parameters like power, pulse duration, and frequency, may influence the efficacy of restoration removal [15]. The retrospective analysis was chosen for this study based on several important considerations. Firstly, there is a scarcity of human subject studies addressing this specific topic in the current literature, with most existing works consisting of case reports or limited case series. Thus, conducting a retrospective analysis would allow us to explore the subject on a larger scale and provide more comprehensive insights.Secondly, the study data were collected from a cohort of a large dental school where restoration removal plays a significant role in dental education. Given that dental students may encounter challenges, particularly in the cementation of all-ceramic restorations, a standardized protocol utilizing Erbium laser was implemented for non-invasive retrieval of miss-cemented restorations in dental student clinics. The retrospective analysis of the clinics with the same retrieval protocol would enhance the reliability of the data collection process and ensure consistency across cases. Lastly, a retrospective analysis is essential in establishing a baseline of clinical data before embarking on a prospective study. This baseline data provides valuable insights into the initial conditions and informs the design and objectives of the future prospective investigation. This retrospective case series aims to present a cohort of successful removals of all-ceramic restorations and prostheses using Erbium laser assistance. The study seeks to provide valuable insights into the optimal laser settings, clinical outcomes, and challenges associated with this approach. By analyzing a range of cases, this research aims to contribute to the understanding of laser-assisted debonding and offer valuable information for clinical practice.

## Materials and methods

### Clinical case series

A retrospective analysis was performed on a cohort of clinical cases involving the removal of prosthetics using laser-assisted techniques. The study examined various aspects of the clinical procedures, including the time required for retrieval, the type and material of the prosthetics, the cement used, laser settings, the success of the removal, cause of removal, and if the prosthesis could be reused. The Human Ethical Committees (Nr KB 962/2022 and HM20027417) granted approvals for the retrospective review and analysis of clinical data. Informed consent was waived by the Institutional Review Board (IRB) as the research protocol involved only the retrospective collection of data and did not entail any changes in the treatment protocol. The data was extracted and analyzed from a de-identified clinical cohort collection. The cases included the use of Er:YAG/Er,Cr:YSGG lasers to remove lithium-disilicate veneers and crowns, zirconium-oxide single crowns, fixed partial dentures (FPDs), and ceramic implant crowns in patients.

## Laser-assisted prosthetic removal protocol

The laser-assisted prosthetic removal procedures were conducted by two experienced clinicians who were certified in laser dentistry (JGD and KGL). The clinical protocol for the procedures was developed based on previous *in vitro* studies. In most cases, local anesthesia was not required for the removal of the prosthetics. However, local anesthesia was administered if cutting of the prosthesis for natural tooth abutments was deemed necessary [7,11–15].

Two types of lasers were used in this clinical series: Er:YAG laser (LightWalker, Fotona, Slovenia) with a tipless handpiece (HO2, Fotona) operating at a power of 2.5–5 W; operation mode QSP/SSP; air/water spray at 2/2 or 6/6; and non-contact mode Er,Cr:YSGG laser (Waterlase, Biolase, USA) with an MX9 Turbo handpiece operating at 5W, 15 PPS; 20 air/20 water spray. The Er:YAG laser parameters varied depending on the type of restoration, with lower settings used for veneers and lithium disilicate crowns, and higher settings for thicker zirconia crowns and fixed partial dentures (FPDs). The specific laser parameters included a range of 170–420 mJ, 12–16 Hz, and operation modes of QSP or SSP.

During the procedure, the laser was directed perpendicular to the surface of the crown from a distance of 5 to 8 mm using a non-contact mode, and the handpiece was moved continuously to ensure even distribution of laser energy. An air/water spray was used throughout the irradiation process. The irradiation was performed on the buccal, lingual, and occlusal surfaces for approximately 30 seconds per surface. For the mesial and distal surfaces, irradiation was achieved at a 45-degree angle from the buccal and lingual aspects of the interproximal space. After approximately 1–3 minutes of irradiation, we attempted to dislodge the prosthesis using a pair of hemostats with digital manipulation. If the prosthesis was not dislodged, the laser irradiation was continued. The removal protocol was standardized based on established methodologies from several previous studies [11–16].

The dislodgment of the prosthesis was assessed after irradiation of all surfaces and periodically during additional irradiation until the prosthesis could be removed using light tapping, elevating forces, and digital manipulation. The removed crowns were minimally cleaned and, in most cases, reused as temporary restorations or re-cemented as permanent restorations in an improved position. Fig 1 provides overall visual illustrations of the prosthesis removal procedure. The prosthesis removal time was meticulously recorded as an integral part of the clinical protocol. The timing commenced from the initial application of laser irradiation and concluded upon the successful retrieval of the prosthesis, aligning with the methodology utilized in our previous *in vitro* studies [11–16].

## Results

### Types of prostheses, luting cements, and removal success

A retrospective review was conducted on twenty-nine patients (eleven males, 18 females) who required debonding of a total of 46 ceramic prostheses with natural tooth or implant abutment, resulting in a total of 52 abutments. Most of the abutments were natural teeth except for 2 implant crowns and one implant-supported FPDs. The retrospective analysis encompassed various clinical procedures, including retrieval time, tooth/implant site, prosthesis type and material, cement type, laser parameters, gender, and age. Detailed information and the included cases can be found in Table 1. Table 2 illustrates the types of cements and prostheses. Majority of the crowns were monolithic CAD/CAM zirconia that were luted with adhesive bonded resin cement (Variolink Esthetics, Ivoclar).

Out of the 52 abutment units, 50 cases achieved successful retrieval without causing damage using either the Er,Cr:YSGG laser (N = 6) or the Er:YAG laser (N = 46). In one case, a crown

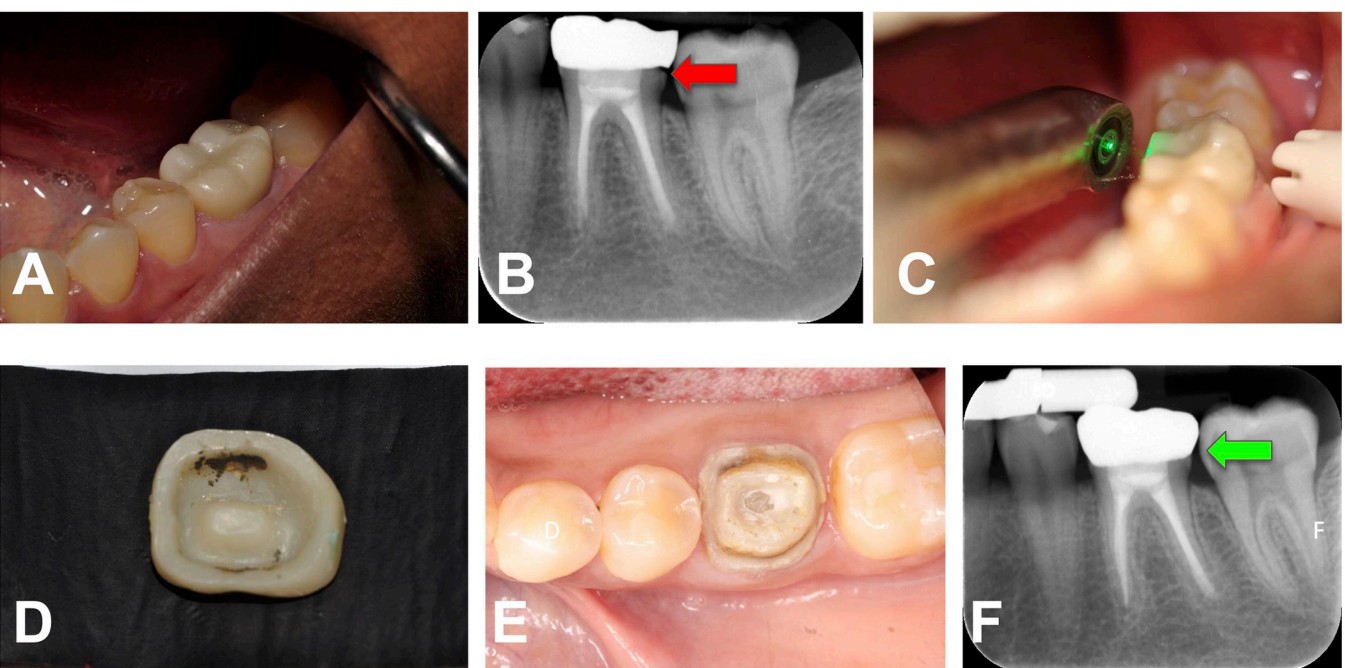

**Fig 1. Debonding and recementation of posterior zIrconia crown on natural tooth abutment.** (A) Existing crown on tooth #19, (B) Periapical radiograph demonstrating an open margin (indicated by red arrow) on the distal due lack of appropriate adjustment of distal crown contact area, (C) Laser-irradiation, (D) Retrieved crown with darkened ablated cement, (E) Abutment tooth after laser-assisted crown removal, and (F) Periapical radiograph demonstrating proper seated definitive crown (indicated by green arrow).

**Table 1. Characteristics of subjects and removed prostheses.**

| Demographic Data | | Prosthesis survival characteristics | | | Time of Removal in minute | |
|---|---|---|---|---|---|---|
| **Ntotal = 29 patients** | | **Reused** | **Remade** | **Damaged during removal** | **Average** | **Standard Deviation** |
| Gender (F/M) | 18/11 | | | | | |
| Age (average/standard deviation) | 45/16 | | | | | |
| Ntotal = 46 prostheses/52 units | | | | | | |
| **Veneers (unit)** | 6 | | 6 | | 2.25 | 0.61 |
| **Crowns (unit)*** | 37 | | | | 6.89 | 8.07 |
| -Lithium disilicate crowns | 7 | 2 | 3 | | 5.86 | 2.41 |
| -Zirconia crowns** | 30 | 17 | 10 | 2 | 7.12 | 8.91 |
| **Fixed partial dentures (unit) *** | 3 | 3 | | | 25 | 10 |

*all natural abutments except for 2 implant titanium abutments, and 1 natural tooth abutment with cast metal post.

**all Er:YAG except for two zirconia crowns were removed using Er,Cr;YSGG.

**1 zirconia crown was removed but had not been replaced.

***3 FPDs, 1 3-unit FPD (2 natural abutment each with 1 pontic), 1 3-unit FPD (2 implant abutment each with 1 pontic) and 1 6-unit FPD (4 implant abutment each and 2 pontics).

***All FPDs were zirconia.

***All 3-unit FPDs were removed using Er:YAG. The 6-unit FPD was removed using Er,Cr;YSGG.

***Time of removal for the FPD was reported per abutment.

**Table 2. Type of cements used.**

| Cement | Veneers | Lithium disilicate crowns | Zirconia crowns | FPD* |
|---|---|---|---|---|
| Bonded resin (Variolink Esthetics, Ivoclar) | 6 | 5 | 20 | |
| Self adhesive resin (Rely X Unicem 2, 3M) | | 1 | 7 | 2 |
| Self adhesive resin (Panavia SA Universal, Kuraray) | | 1 | 1 | |
| Zinc Oxide Eugenol (Flynal Permanent ZOE Cement, Dentsply Sinrona) | | | 1 | |
| Unknown | | | 1 | 1 |

was partially sectioned to preserve the adhesive strength between the post and tooth, and another crown fractured during the debonding process. The restorations included 13 lithium disilicate (6 veneers and 7 crowns) and 33 zirconia (30 crowns and 3 FPDs) prostheses. The removal process is visually depicted in Figs 1 to 8.

Among the restorations, 44 were removed from natural teeth (Figs 1 to 5), six were removed from metal implant abutments, and two were removed from zirconia implant abutments (Figs 2 and 7). Successful debonding was achieved using either an Er:YAG laser (N = 46) or an Er,Cr:YSGG laser (N = 6). The zirconia prostheses included 30 single crowns with natural tooth abutments, one 3-unit FPD with natural teeth abutments, and two implant-supported FPDs (Figs 1, 2, 5 and 6). Most cases involved in-house fabricated restorations, and the cements used predominantly consisted of self-adhesive resin dual-cure cements, with one case utilizing zinc oxide eugenol provisional cement.

## Removal time

Table 1 presents the average values and standard deviations for different types of prostheses, abutments, and cements. The prosthetic retrieval times for lithium disilicate restorations varied depending on the type, ranging from 90 seconds for a veneer to 4–10 minutes for a crown.

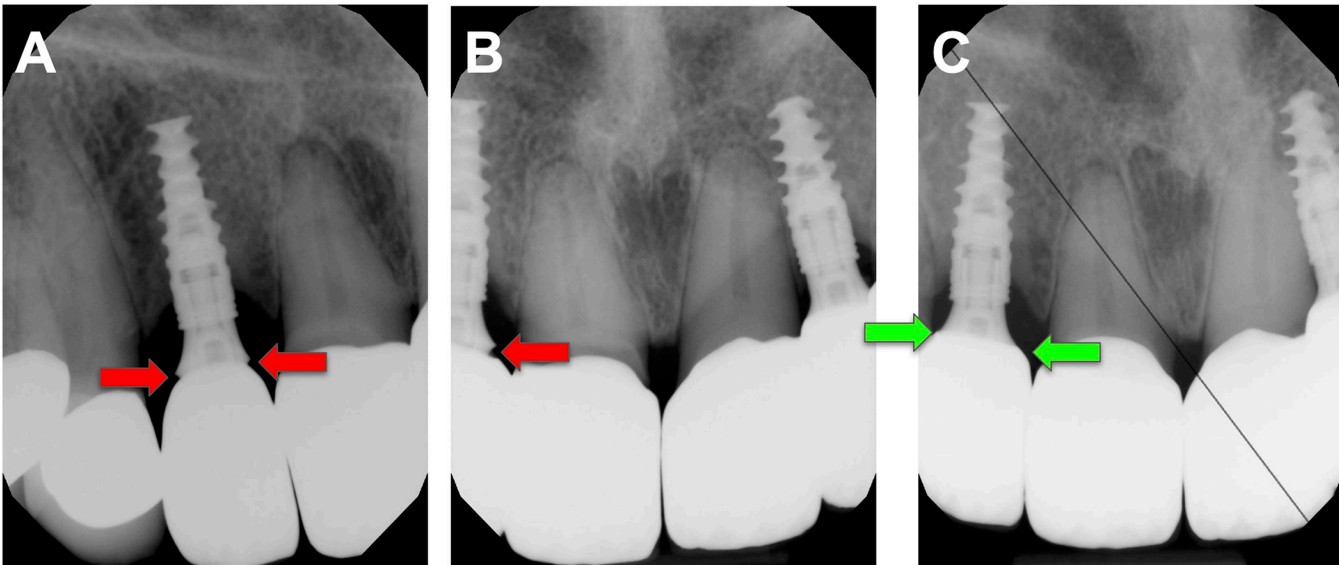

**Fig 2. Debonding and re-cementation of anterior zIrconia crown on implant abutment.** (A) and (B) Periapical radiographs demonstrating #7 zirconia crown was missed seated during the cement due to insufficient interproximal adjustment, red arrows indicating the open unseated mesial and distal margin, (C) Periapical radiograph taken after successful laser-assisted crown retrieval and existing crown re-cementation, green arrows indicating appropriate seating of the crown.

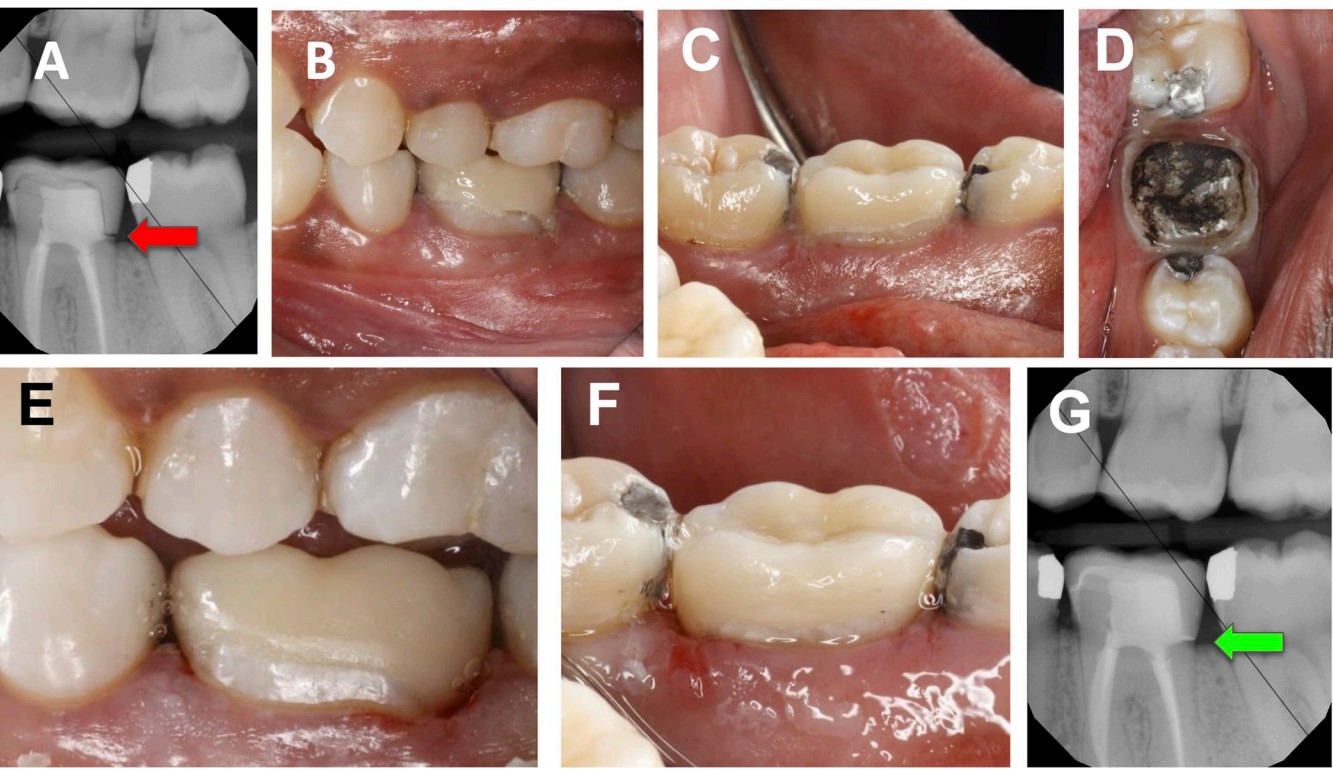

**Fig 3. Debonding and re-cementation of posterior lithium disilicate crown on natural tooth abutment.** (A) Bitewing radiograph indicating improper seating with open distal margin shown by red arrow, (B) and (C) Buccal and lingual preoperative views, (D) abutment tooth immediately after crown removal demonstrating darkened ablated residual cement, (E) and (F) Buccal and lingual view of re-cemented crown, and (G) Bitewing radiograph indicating appropriate seating with sealed distal margin shown by green arrow.

In contrast, zirconia single crowns exhibited a wider range, requiring 2 to 30 minutes for removal. The FPDs, all made of zirconia, had the longest removal time overall, ranging from 10 to 35 minutes per unit.

The duration for debonding procedures varied greatly based on factors such as the restoration type, ceramic material type and thickness, cement type, and the fit of the abutment and restoration. Removing a zirconia crown took more time compared to a lithium disilicate veneer or crown for a single restoration. Additionally, the removal of a fixed partial denture required more time compared to a single crown. Visual representations of crown removal can be found in Figs 1,2,3,4 and 7.

## Reasons for prosthesis removal

Table 3 presents the different reasons for prosthesis removal. These include esthetic concerns, improper seating, open margin detection, caries, gingival inflammation, fracture, undercontoured restoration, implant abutment fracture, and the need for access for peri-implantitis treatment. Figs 1–5, illustrate examples of defective (open) margins, incorrect due to incomplete or incorrect seating of the prosthesis during the cementation process. Fig 6 demonstrated retrieval of implant prosthesis with fractures of zirconia implant abutments. Fig 7 demonstrated a removal of an implant crown with an open margin. Fig 8 demonstrated retrieval of implant prosthesis with the need for access to implant fixtures for peri-implantitis treatment.

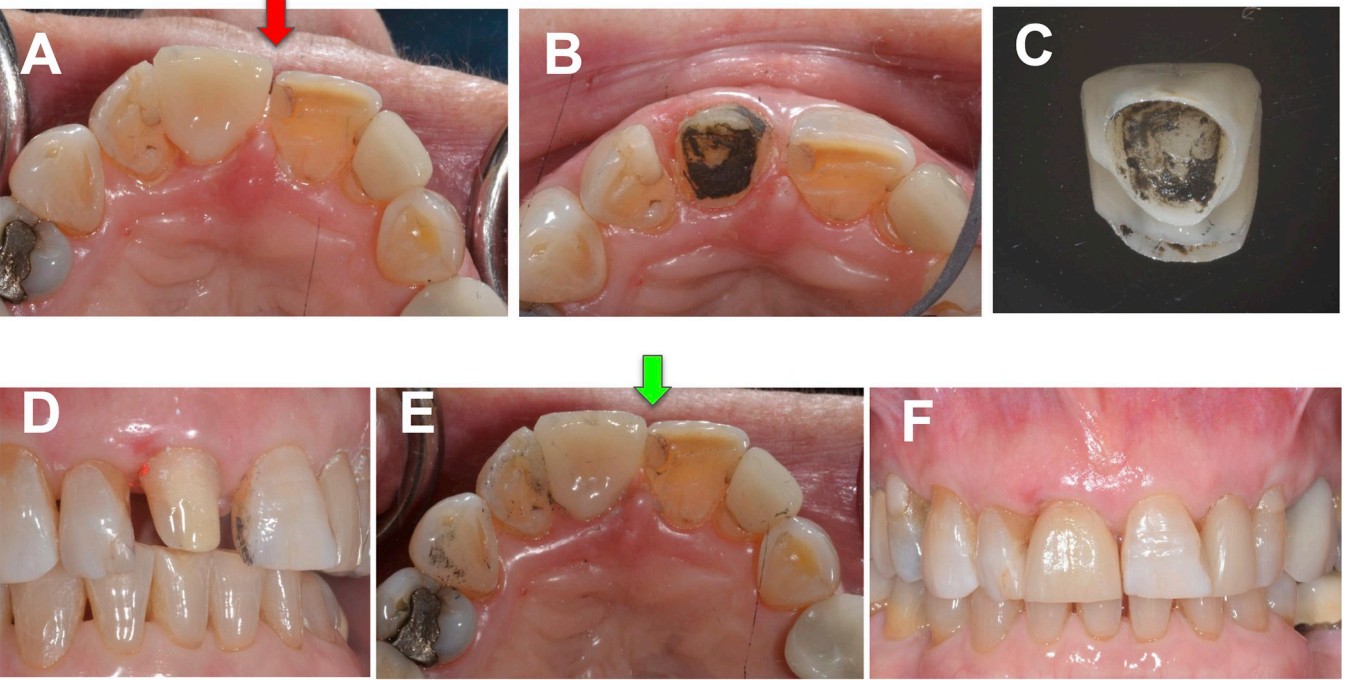

**Fig 4. Debonding and re-cementation of anterior lithium disilicate crown on natural tooth abutment.** (A) Existing recently cemented #8 crown was wrongly positioned during the cementation process resulting in the mesial diastema see red arrow, (B) Abutment tooth after Er:YAG laser debonding, (C) Retrieved intact crown with ablated cement remaining, (D) Abutment after cement removal, (E) and (F) Crown in place after re-cementation.

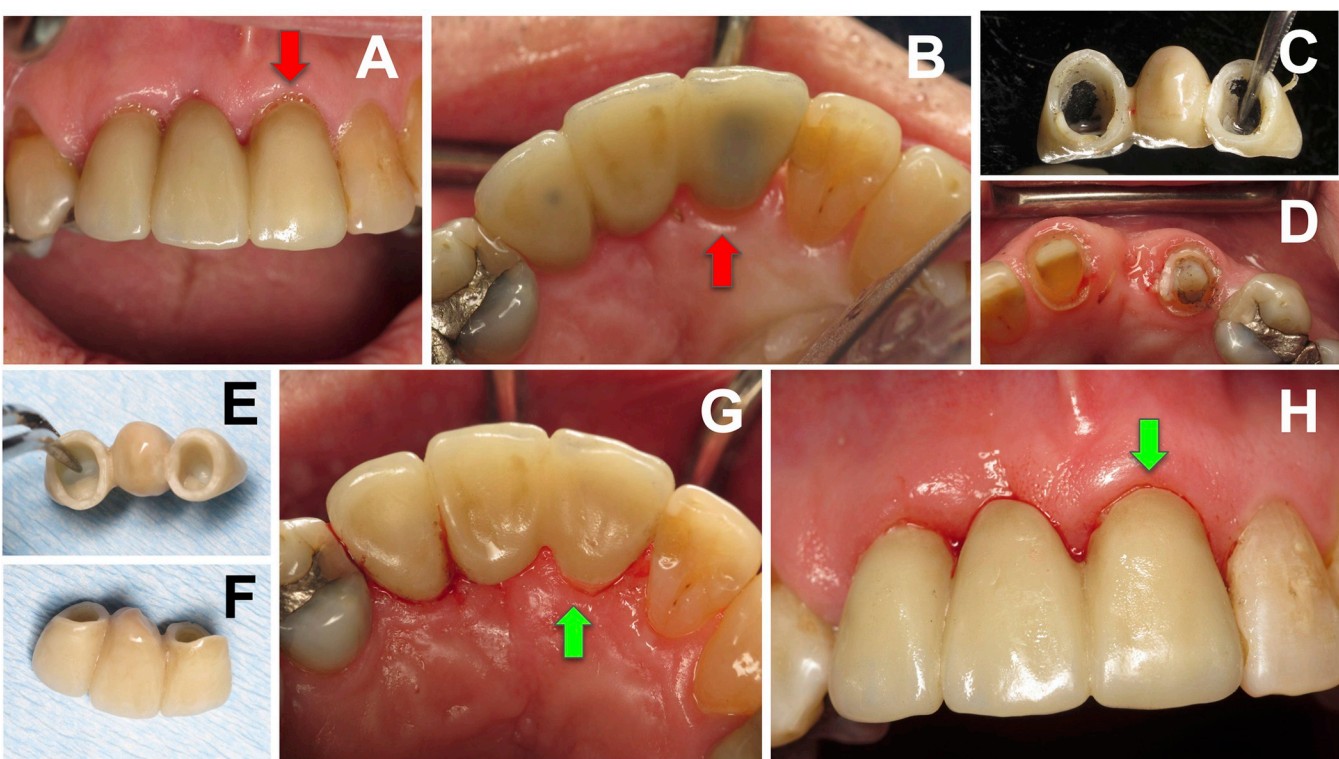

**Fig 5. Debonding and re-cementation of zirconia FPD on natural tooth abutments.** (A) and (B) FPD was improperly seated during cementation, note the open margin facially and lingually shown by red arrows, (C) and (D) FPD and abutments immediately after Er:YAG debonding, (E) and (F) FPD was air-abraded and steam cleaned, and (G) and (H) Re-cemented FPD with closed margin, see green arrows.

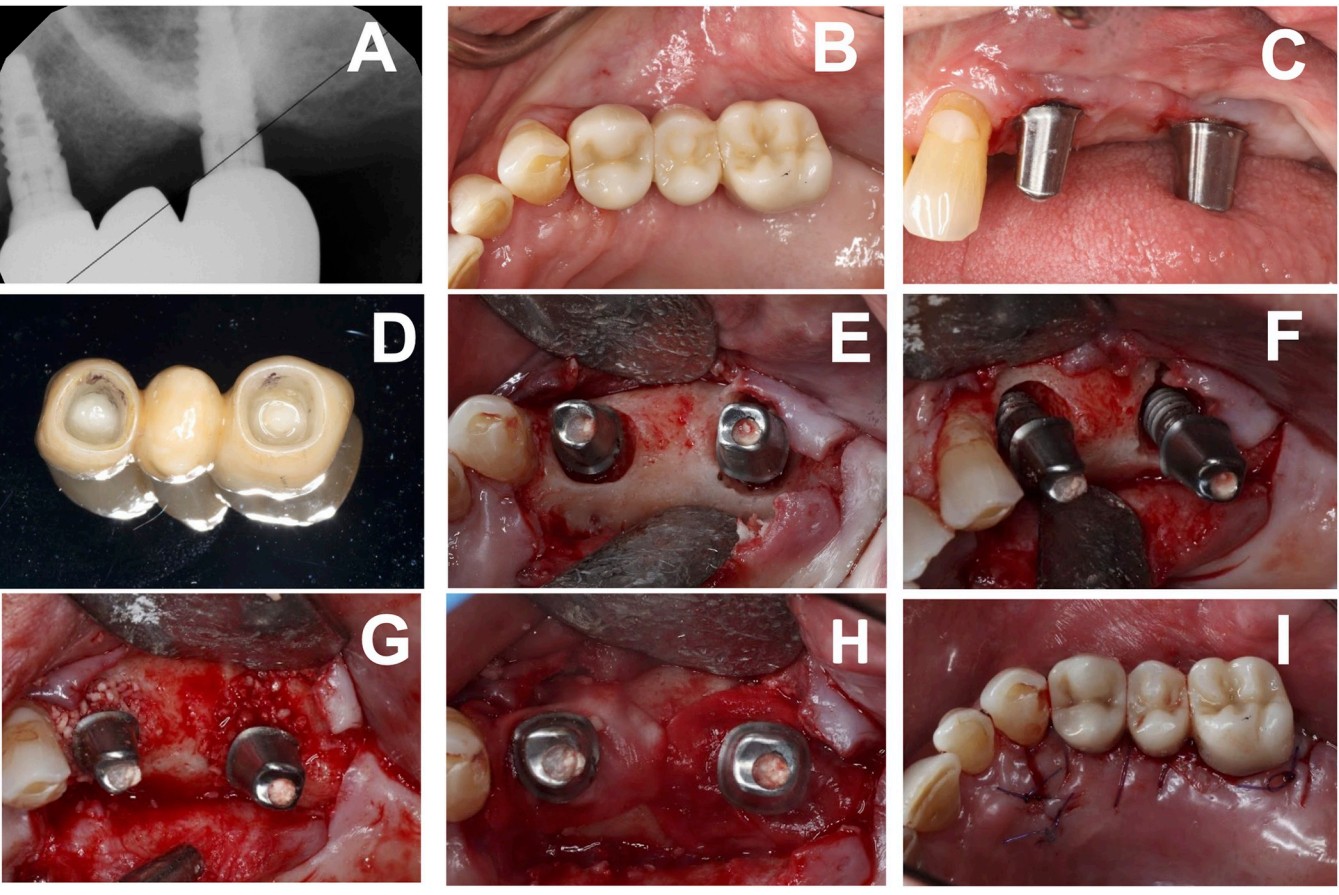

**Fig 6. Retrieval of cement-retained implant-supported zirconia FPD.** (A) Periapical radiograph demonstrating one peri-implantitis bone loss of #12 and 14 implants, (B) Preoperative FPD before debonding, (C) Titanium abutments after debonding, D) FPD after debonding, being air-abraded, and steam cleaned, (E) flap opening and removal of peri-implantitis tissue, (F) Bone decortication, (G) and (H) grafting and membrane in place, (I) FPD re-cementation after suturing.

### Type of the erbium laser and settings

Table 4 illustrates the laser types and settings employed by practitioners based on the type of prosthesis. In all cases, erbium lasers were utilized, with six units debonded using the Er,Cr: YSGG laser and 46 units debonded using the Er:YAG laser. Successful removal of all-ceramic crowns from both teeth and implant abutments was achieved using erbium lasers. While both erbium lasers were effective in removing zirconia crowns, the Er:YAG laser exhibited higher debonding efficiency. The only zirconia crown that required partial sectioning for removal was debonded using the Er,Cr:YSGG laser. The power settings for debonding ranged between 4–5 W, depending on the type of restoration. Lower power (4.5 W) was used for lithium disilicate crowns, while higher power (5 W) was necessary for removing zirconia restorations, particularly FPDs. Table 4 provides a detailed description of the laser settings for each case. Notably, practitioner KLG utilized lower laser setting values compared to practitioner JGD.

### Recementation of irradiated ceramic restorations

Table 1 provides information on the reusability and integrity of the retrieved prostheses. The majority of debonded prostheses (44 out of 46) remained intact, representing a success

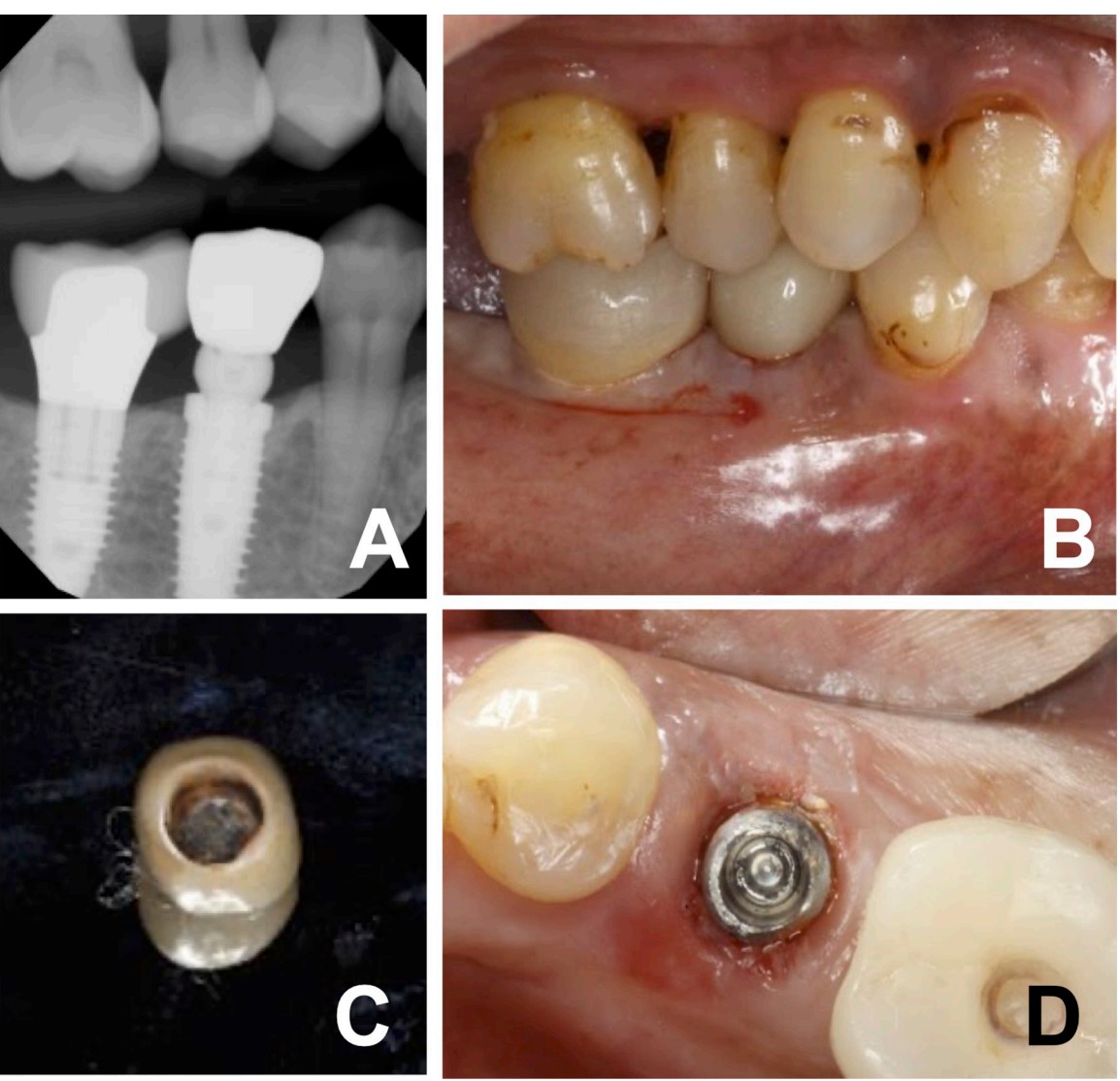

**Fig 7. Removal of cement-retained zirconia implant crown.** (A) Bitewing radiograph demonstrating miss-seating #29 crown, (B) Preoperative buccal view, (C) Intaglio surface of the crown immediately after removal, and (D) Implant abutment after crown removal.

debonding rate of 95.65%. Only 2 out of 46 prostheses (4.35%) were fractured after debonding. In one case, a crown required partial sectioning on the facial surface to examine the interface between the tooth and custom metal post/core. No damage was observed on the underlying abutment in any of the cases. The only restoration that needed partial sectioning was a zirconia crown cemented to a metal custom post/core. Macroscopic examination at magnifications of 2.5–3.5 revealed no structural damages or cracks on the debonded ceramic restorations, regardless of the duration of laser irradiation, type of erbium laser used, or type of ceramic material. Out of the 46 retrieved prostheses, only 22 (19 crowns and 3 fixed partial dentures) were successfully recemented during the same visit as debonding, representing a recementation rate of 47.83% (Fig 4). Six restorations that were debonded, were used as provisional restorations until new prostheses were fabricated.

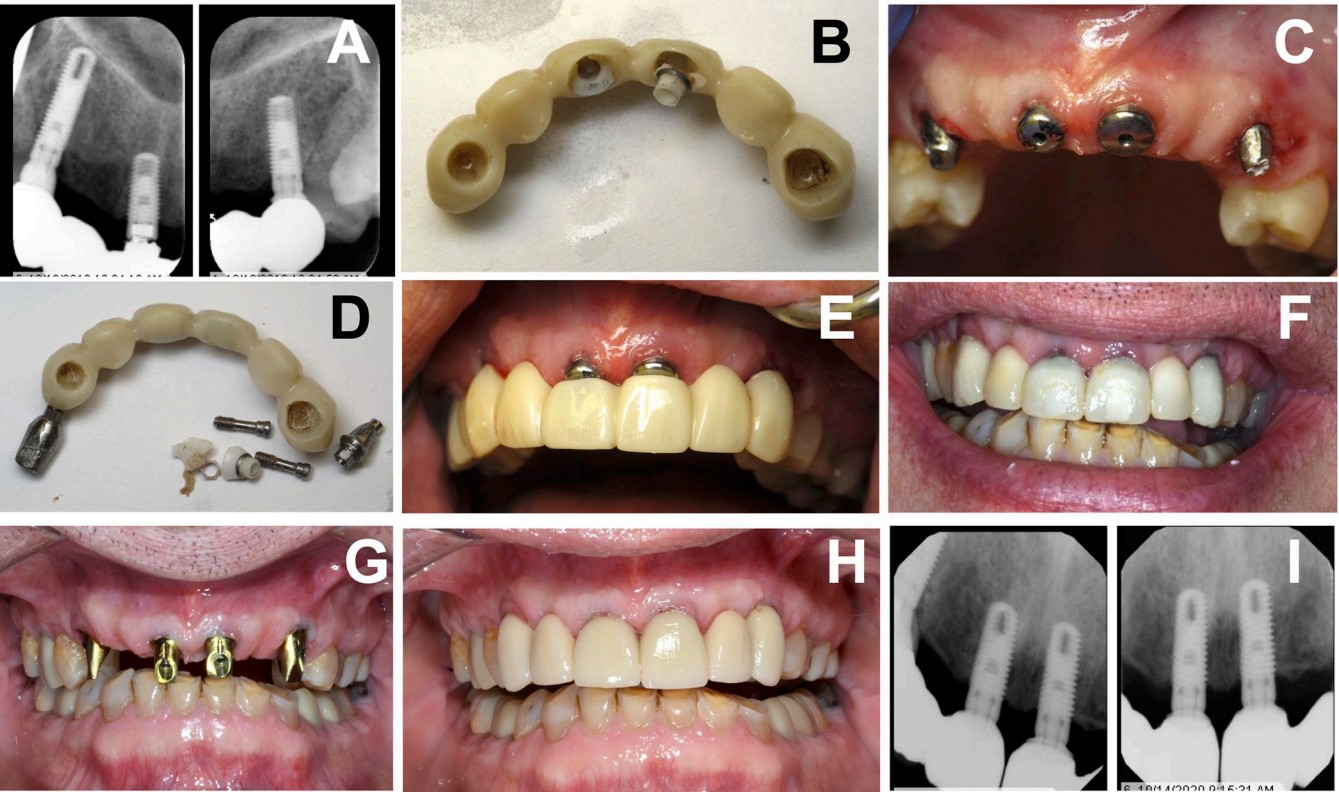

**Fig 8. Removal of Implant-supported cement-retained FPD with fractured zirconia abutments.** (A) Periapical radiographs demonstrating fracture of zirconia abutments #8 and 9, (B) Debonded FPD, (C) Healing abutment was placed on #8 and 9 implants, (D) composite resin was added to convert existing FPD to be used as provisional FPD, (E) Converted provisional FPD in place, (F) Postoperative image after a few weeks of removal, (G) New definitive custom titanium abutments, (H) New definitive prosthesis in place, and (G) Postoperative periapical radiographs.

## Discussion

This retrospective clinical series represents one of the most extensive and comprehensive analyses conducted to date on the debonding of ceramic restorations using Erbium lasers in real-life clinical settings. It boasts a large sample size, making it one of the largest cohorts of laser-assisted ceramic fixed prosthesis removal. By encompassing the full spectrum of cases, our

**Table 3. Causes of prosthetic removal.**

| Causes of removal | Veneers | Crowns | Lithium disilicate crowns | Zirconia crowns | Fixed partial dentures |
|---|---|---|---|---|---|
| Esthetics | 5 | 16 | 1 | 15 | |
| Seating | | 10 | 3 | 4 | 1 |
| Open margin detection* | | 3 | 2 | 1 | |
| Caries | | 4 | 1 | 3 | |
| Gingival inflammation | | | | 2 | |
| Fracture | 1 | | | | |
| Undercontoured restoration | | | | 1 | |
| Implant Abutment Fracture | | | | | 1 |
| Access for peri-implantitis treatment | | | | | 1 |

*open margin without inappropriate seating.

**Table 4. Practitioners, lasers and settings, and types of debonded prostheses.**

| Practitioner | Laser | Setting | Veneers | Lithium disilicate crowns | Zirconia crowns | FPD* |
|---|---|---|---|---|---|---|
| KGL | Er:YAG | 170mJ; 15Hz; 6W;6A, SSP | 5 | 3 | 17 | |
| | Er:YAG | 200mJ, 15Hz, 3W; 6W;6A, QSP | | | 3 | |
| JGD | Er:YAG | 245 mJ, 16 Hz, 4 W, 2W;2A, SSP | 1 | | | |
| | Er:YAG | 200mJ, 15Hz, 3W; 6W;6A, QSP | | | | |
| | Er:YAG | 300 mJ, 15Hz, 4.5W, 2W;2A, SSP | | 4 | 5 | |
| | Er:YAG | 325 mJ,15 Hz, 4.8W, 2W;2A, SSP | | | 1 | |
| | Er:YAG | 300–325 mJ, 15Hz, 5W, 2W;2A, SSP/QSP | | | | 1 (2) |
| | Er:YAG | 335 mJ, 15Hz, 5W, 2W;2A, SSP | | | 2 | |
| | Er:YAG | 420 mJ, 12Hz, 5W, 5W;5A, QSP | | | | 1 (2) |
| | Er,Cr;YSGG | 5 Watt, 15 Hz, 20W;20A | | | 2 | 1 (4) |

* Number of prostheses (number of abutments).

study seeks to provide a comprehensive overview and valuable insights into the clinical outcomes associated with Erbium laser removal of all-ceramic restorations/prostheses. The inclusion of diverse cases will contribute to a more robust understanding of the subject matter and pave the way for future investigations to target specific subgroups based on refined selection criteria.The study demonstrates a remarkable success rate of over 95% for intact debonding of ceramic prostheses using Erbium lasers. The average time required for prosthesis removal varied depending on the type of restoration. Veneers could be removed in as little as 2.25 ±0.61 minutes, crowns took an average of 6.89 ±8.07 minutes, and FPDs required approximately 25 ±10 minutes per abutment. When comparing different crown materials, lithium disilicate crowns were generally faster to remove with an average time of 5.86 ±2.41 minutes, compared to zirconia crowns which took around 7.12±8.91 minutes. The majority of luting cements used were bonded or self-adhesive, while monolithic zirconia crowns constituted the majority of prostheses examined. The most common reasons for prosthetic removal were esthetic concerns, accounting for 34.78% of cases, followed by miss-seating issues at 23.91%. The Er:YAG laser, particularly the Light Walker by Fotona, was the most frequently used laser and exhibited versatility in handling various types of ceramic restorations, abutments, and luting cements. However, only 47.83% of the retrieved prostheses were considered suitable for re-cementation at the same visit, indicating that over half of the prostheses required replacement.

Multiple studies have indicated that erbium lasers are a reliable and predictable method for debonding all-ceramic restorations without causing damage to the underlying teeth or implant abutments. When examined under magnification, the ceramic surfaces typically show no or minimal alterations [11–14,16,23,24]. The emission of light energy from erbium lasers can penetrate through the ceramic materials, and this transmission is influenced by various factors. These include the surface area of the restoration [11–14,16,23,24], the translucent properties of the ceramic material [22], the thickness of the restoration [22,25], and the composition of the cement used [26]. The bond between the ceramic restoration and the tooth or implant abutment is primarily disrupted within the cement layer and at the interface between the ceramic and the cement. The success of prosthesis removal depends on several factors, including the type of restoration and its material, the properties of the cement used, the surface area of the crown-abutment interface, the operator's experience and level of expertise, and the location of the restoration in the oral cavity [15].

The *in-vivo* time required for laser irradiation to achieve prosthesis removal was found to be longer compared to observations made in *ex-vivo* conditions for single crowns [12,13,15].

Debonding veneers required a very short time compared to individual crowns, which displayed a wider range of debonding times. While some crowns could be removed within a few minutes, similar to what has been reported in *ex-vivo* conditions [12,13,15], others required a longer irradiation time. For the removal of FPDs, significantly longer irradiation times were noted, but there is a lack of previously published data for direct comparison. *ex-vivo* simulated studies can provide valuable guidance in selecting laser parameters. However, in clinical applications, manipulating the laser handpiece for optimal access and efficiency presents a challenge and increases working time. Interruptions to reposition the mirror and handpiece, inspect the area, obtain feedback from the patient, and enable indirect vision are more common in the *in-vivo* setting. Additionally, angulation of the laser handpiece can be challenging in interproximal sites compared to the *ex-vivo* model. Laser irradiation tends to favor the buccal and some lingual aspects due to easier access, while ideally, it should be evenly distributed over the entire surface. Thicker interproximal areas of FPDs also require longer irradiation. The observed retrieval time in this clinical series is thus higher than what has been suggested by *ex-vivo* studies [12,13,15]. Retention and resistant forms of the prosthetic abutments can further hinder removal, as the ablated cement does not undergo volumetric changes despite losing its adhesive properties. The cement filling the gap between the crown and abutment surface can create friction, making it challenging to remove the prosthesis without applying elevating or tapping force. There is a learning curve in recognizing when the cement has lost its adhesive properties, and the restoration remains retained on the tooth solely by mechanical/physical properties.

The majority of crowns and FPDs that were retrieved due to seating problems were cleaned (with appropriate intaglio surface treatments) and re-cemented onto the abutment teeth or implants in an improved position during the same appointment, as shown in Figs 2 to 6. In cases where the retrieved restorations were found to be deficient and new prostheses were required, the retrieved restorations could occasionally be used as provisional restorations while the replacements were being fabricated. Note that one crown had to be partially sectioned because there were concerns that laser irradiation might impact the adhesive strength of the cement between the dentin and the subgingivally placed custom post/core, which could jeopardize its long-term prognosis. Once the irradiation could be directed away from the tooth-core interface, the crown was successfully removed through debonding. However, it was determined that the crown was not suitable for reuse. It is recommended that future research investigates the effect of laser irradiation during crown debonding on the adhesive properties of resin-bonded cores and posts. Understanding this aspect will provide valuable insights for improving the overall success and prognosis of such restorations.

The efficiency of debonding ceramic restorations can be influenced by the composition of the cement used, as different cements can have varying ablation thresholds and volume loss during laser irradiation. This variation is likely due to differences in the amount of water molecules and unpolymerized monomers present in the cements. It has been shown that cements such as Rely X Unicem U100 (3M-ESPE), Variolink II (Ivoclar), and Panavia F-20 (Kuraray Dental) exhibit lower volume losses during debonding [25]. In this clinical series, the majority of units were cemented using Rely X Unicem 2 (3M-ESPE) and Variolink Esthetics (Ivoclar) cements, which may have contributed to longer debonding times compared to other cements. The differences in debonding time could also be attributed to slight variations in laser settings or the properties of the materials used in the cements and ceramics. Additionally, bonded restorations may generally be more challenging to remove compared to non-bonded restorations due to the adhesive properties of the cement. These factors highlight the importance of considering the specific properties of the cement used and its interaction with the ceramic materials when performing debonding procedures.

The transmission ratio through ceramic materials decreases as the thickness of the ceramic increases [22]. This observation is consistent with clinical experience, which has shown that thinner ceramics have a higher transmission ratio. The time required to debond a restoration is positively correlated with the surface area and volume of the tooth and crown, as well as the volume of the luting cement [27]. The type of ceramic and the thickness of the restoration strongly influence the transmission of laser energy and, consequently, the efficiency of debonding. Debonding a thin veneer or a lithium disilicate crown takes less time, which is in line with findings from in vitro studies [25,27–30]. Our clinical observations are consistent with previous research that has reported longer debonding times for zirconia crowns compared to lithium disilicate crowns when removing them from natural teeth [11], as well as from zirconia or titanium abutments [12–14]. These findings highlight the importance of considering the ceramic type and thickness when estimating the time required for debonding, as they significantly impact the transmission of laser energy and, consequently, the efficiency of the debonding process.

The laser settings used in these clinical cases were slightly lower compared to those in *ex-vivo* studies. Different laser settings have been reported for the removal of prosthetic restorations. In these clinical cases, conservative laser settings were employed due to limited available in-vivo references and the generally thicker and variable crown thickness [15,23,31]. A recent scoping review highlighted the consistency in similar applications of laser settings for both Er: YAG and Er,Cr:YSGG lasers [15]. While both erbium lasers are effective in debonding restorations, cases treated with the Er,Cr:YSGG laser took longer due to its lower absorption coefficients compared to the Er:YAG laser [23,31]. The wavelength of the Er,Cr:YSGG laser penetrates deeper into the tissue and requires more time to reach the evaporation temperature, while the substance heated by the Er:YAG laser reaches ablation temperatures faster [32]. However, the two erbium lasers can be used interchangeably for debonding ceramic restorations [15,18,23,31]. The optimal laser power setting in the reported cases ranged from 3.0 to 5.0 W, depending on the restoration. The higher efficiency of the Er:YAG laser observed in these cases aligns with previously reported observations [14,20,31,33]. For crown removal, a laser setting of at least 3.5 to 4 W of power with a 25 Hz pulse rate is recommended.

To ensure the safety of pulpal tissue, periodontium, and bone, it is important to limit the temperature increase caused by erbium lasers to a maximum of 5.5˚C [34–36], as exceeding this threshold can lead to thermal injury. Both erbium lasers used in these cases provide continuous irrigation during irradiation, effectively minimizing temperature changes in adjacent vital tissues and keeping them within the safe range specified by the parameters used. In this reported cases, minimal or no local anesthesia was required, and no pulpal symptoms were experienced during or after the debonding procedure. These findings suggest that the laser irradiation using the proposed settings resulted in minimal temperature increases, indicating that laser-assisted ceramic restoration removal does not pose a thermal risk to the tooth, implant, adjacent tissues, or cause damage to the ceramic restorative material. Patients showed high compliance, comfort, and acceptance towards the laser-assisted removal of restorations. In most cases, local anesthesia was not necessary as the laser irradiation effectively released the bond of the cement, allowing for gentle removal of the restorations from the abutments. It is crucial for dental professionals to have adequate training and knowledge regarding laser safety and parameters, especially when dealing with different tissues and ceramics, to prevent any damage to the natural tooth abutment. Further clinical studies are needed to enhance efficiency and optimize laser parameters and procedure efficiency, while ensuring the safety of the parameters for practical clinical application.

The current study has certain limitations that should be acknowledged. Firstly, it is a retrospective clinical series, which may have inherent biases and limitations associated with its

design. Secondly, the sample size and practitioners were limited, which may affect the generalizability of the findings. Third the study primarily focused on the debonding of ceramic restorations using erbium lasers, and other types of lasers were not explored. There was a lack of comparison between the two Erbium lasers. Fourth, the presence of various types of ceramic compositions, materials, and levels of translucency could have influenced laser transmission, potentially impacting the retrievability and retrieval time of the restorations [11–16,37]. Finally, there was no standardization of clinical protocols in removal of different types of prostheses. To further advance our understanding in this field, future research should include well-designed controlled clinical trials and longitudinal prospective studies. These studies can provide more robust evidence on the safety and efficacy of different laser parameters for irradiation of ceramic restorations with varying thicknesses and luted with cements of diverse chemical compositions. Moreover, there is a need to develop standardized protocols for laser settings in the removal of restorative appliances. These efforts will contribute to the establishment of best practices and guidelines for laser-assisted restoration removal, enhancing the quality of dental care and patient outcomes.

## Conclusion

Laser-assisted ceramic restoration removal using Erbium lasers provides a reliable and predictable treatment approach. It offers a safe and effective method to retrieve ceramic restorations/appliances from both natural teeth and implant abutments without causing damage to the restorative material or the abutment surface.

## Author Contributions

**Conceptualization:** Janina Golob Deeb, Kinga Grzech-Lesniak, Sompop Bencharit.

**Data curation:** Janina Golob Deeb, Kinga Grzech-Lesniak, Sompop Bencharit.

**Formal analysis:** Janina Golob Deeb, Kinga Grzech-Lesniak, Sompop Bencharit.

**Investigation:** Janina Golob Deeb, Kinga Grzech-Lesniak, Sompop Bencharit.

**Methodology:** Janina Golob Deeb, Kinga Grzech-Lesniak, Sompop Bencharit.

**Project administration:** Janina Golob Deeb.

**Resources:** Janina Golob Deeb.

**Supervision:** Sompop Bencharit.

**Validation:** Janina Golob Deeb, Kinga Grzech-Lesniak, Sompop Bencharit.

**Visualization:** Janina Golob Deeb, Kinga Grzech-Lesniak, Sompop Bencharit.

**Writing – original draft:** Janina Golob Deeb, Sompop Bencharit.

**Writing – review & editing:** Janina Golob Deeb, Kinga Grzech-Lesniak, Sompop Bencharit.

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
