## [Decision Letter · Decision Letter 0]

11 Jul 2023

PONE-D-23-15529Evaluation of the Effectiveness and Practicality of Erbium Lasers for Ceramic Restoration Removal: A Retrospective Clinical AnalysisPLOS ONE

Dear Dr. Bencharit,

Thank you for submitting your manuscript to PLOS ONE. After careful consideration, we feel that it has merit but does not fully meet PLOS ONE’s publication criteria as it currently stands. Therefore, we invite you to submit a revised version of the manuscript that addresses the points raised during the review process.

We look forward to receiving your revised manuscript.

Kind regards,

Mirza Rustum Baig

Academic Editor

PLOS ONE

3. We note that Figures 1-8 in your submission contain copyrighted images. All PLOS content is published under the Creative Commons Attribution License (CC BY 4.0), which means that the manuscript, images, and Supporting Information files will be freely available online, and any third party is permitted to access, download, copy, distribute, and use these materials in any way, even commercially, with proper attribution. For more information, see our copyright guidelines: http://journals.plos.org/plosone/s/licenses-and-copyright.

a. You may seek permission from the original copyright holder of Figures 1-8 to publish the content specifically under the CC BY 4.0 license.

Additional Editor Comments:

Both the reviewers found the study pertinent and interesting. However, there are some concerns about the study design and the numbers of different types of restorations reported which reduces the effectiveness of the outcomes.

Nevertheless, the preliminary findings are valuable and may warrant publication.

Please address the concerns raised by the reviewers.

Reviewers' comments:

Reviewer's Responses to Questions

**Comments to the Author**

1. Is the manuscript technically sound, and do the data support the conclusions?

Reviewer #1: Partly

Reviewer #2: Yes

2. Has the statistical analysis been performed appropriately and rigorously? 

Reviewer #1: Yes

Reviewer #2: Yes

3. Have the authors made all data underlying the findings in their manuscript fully available?

Reviewer #1: Yes

Reviewer #2: Yes

4. Is the manuscript presented in an intelligible fashion and written in standard English?

Reviewer #1: Yes

Reviewer #2: Yes

5. Review Comments to the Author

Reviewer #1: I appreciate the authors thoughts. I have the following concerns.

1. Retrospective design for this study is debatable. Kindly justify the use of this design

2. The study should have standardization in inclusion criteria, like type of crown, location of crown, type of prosthesis like FDP, laminate or implant. All cannot be included in one study design. It can lead to many confounding variables and the impact is reduced.

3.Burning/ debonding with lasers are done, but how was the crown removed. It is done either done by conventional crown removers or with hand removal. The results denote for few crowns the time taken is more.It denotes the conventional removal was supported in removal. The whole study can become arguable.

Reviewer #2: This clinical study evaluated the effectiveness of erbium laser in the removal of ceramic crowns.

The research is valuable as it is an in vivo study, and the data has been presented well.

This reviewer has a minor recommendation: The study included ceramic restorations with different light transmission properties. It would be beneficial to discuss the findings of a recent study that includes zirconia ceramic materials with different translucency properties: Birand, Cemil, and Sevcan Kurtulmus-Yilmaz. "Evaluation of Er, Cr: YSGG laser irradiation for debonding of zirconia hybrid abutment crowns from titanium bases." Lasers in Medical Science 37.6 (2022): 2675-2685.

6. PLOS authors have the option to publish the peer review history of their article (what does this mean?). If published, this will include your full peer review and any attached files.

Reviewer #1: No

Reviewer #2: No

---

## [Author Response · Author response to Decision Letter 0]

25 Jul 2023

RESPONSES TO COMMENTS FROM EDITORS AND REVIEWERS

EDITORS:

RESPONSE: We appreciate the comment.

TEXT CHANGE: The manuscript was reformatted according to the guideline provided.

RESPONSE: Thank you for providing additional information. Given that the study was a retrospective chart review and there were no changes in the treatment protocol, and no minor subjects were involved, it is reasonable that the Institutional Review Board (IRB) waived the need for informed consent. The approval from the Human Ethical Committees (Nr KB 962/2022 and HM20027417) allows for the retrospective review and analysis of the clinical data without requiring additional informed consent. This approach ensures compliance with ethical standards while maintaining patient confidentiality and privacy.

TEXT CHANGE: The following information was amended in the Methods.

“The Human Ethical Committees (Nr KB 962/2022 and HM20027417) granted approvals for the retrospective review and analysis of clinical data. Informed consent was waived by the Institutional Review Board (IRB) as the research protocol involved only the retrospective collection of data and did not entail any changes in the treatment protocol.”

3. We note that Figures 1-8 in your submission contain copyrighted images. All PLOS content is published under the Creative Commons Attribution License (CC BY 4.0), which means that the manuscript, images, and Supporting Information files will be freely available online, and any third party is permitted to access, download, copy, distribute, and use these materials in any way, even commercially, with proper attribution. For more information, see our copyright guidelines: http://journals.plos.org/plosone/s/licenses-and-copyright.

a. You may seek permission from the original copyright holder of Figures 1-8 to publish the content specifically under the CC BY 4.0 license.

RESPONSE: Thank you for providing that information. If all photos used in the study were taken by the authors and have not been published previously, then the authors hold the copyright to the images. Since the images are original and unpublished, there is no need for external permission to publish them in the study.

TEXT CHANGE: N/A.

RESPONSE: Thank you for the valuable information. The reference list has been meticulously reviewed and thoroughly validated to ensure the precise alignment with the respective citations.

TEXT CHANGE: N/A.  Additional Editor Comments:  Both the reviewers found the study pertinent and interesting. However, there are some concerns about the study design and the numbers of different types of restorations reported which reduces the effectiveness of the outcomes. Nevertheless, the preliminary findings are valuable and may warrant publication. Please address the concerns raised by the reviewers.  RESPONSE: We express our sincere gratitude to the Editor and Reviewers for their valuable suggestions and insightful comments. We have diligently considered each recommendation and made the necessary amendments to the manuscript accordingly. Your constructive feedback has significantly contributed to improving the quality and clarity of this work. Once again, we appreciate your thoughtful evaluation and are committed to ensuring that the revised manuscript meets the highest scholarly standards.

TEXT CHANGE: Please see responses to the reviewers below.

  REVIEWER #1:

I appreciate the authors thoughts. I have the following concerns. 1. Retrospective design for this study is debatable. Kindly justify the use of this design

RESPONSE: Thank you for raising this important question. The retrospective design of this study is based on three main rationales. First, there are currently no human subject studies available on this specific topic in the literature. The existing literature primarily consists of case reports or a limited number of case series, which necessitates further investigation through a larger-scale study.

Second, the data for this study were collected from a cohort of a large dental school where restoration removal is a crucial aspect of dental education. Given that dental students may encounter challenges, particularly in the cementation of all-ceramic restorations, a standardized protocol utilizing Erbium laser was implemented for non-invasive retrieval of miss-cemented restorations in dental student clinics. This protocol ensures consistency and enhances the reliability of the data collection process.

Lastly, a retrospective approach was deemed essential to establish a baseline of clinical data before embarking on a prospective study. This baseline data provides valuable insights into the initial conditions and helps inform the design and objectives of the future prospective investigation.

By employing a retrospective design, we aimed to address these rationales and contribute meaningful findings to the existing literature. The study's methodology was carefully developed to ensure accurate data collection and analysis while minimizing potential biases. We appreciate your thoughtful consideration and have incorporated these justifications into the manuscript to strengthen the research rationale.

TEXT CHANGE: The rationale for retrospective analysis was added in the end of the introduction as follows.

“The retrospective analysis was chosen for this study based on several important considerations. Firstly, there is a scarcity of human subject studies addressing this specific topic in the current literature, with most existing works consisting of case reports or limited case series. Thus, conducting a retrospective analysis would allow us to explore the subject on a larger scale and provide more comprehensive insights.Secondly, the study data were collected from a cohort of a large dental school where restoration removal plays a significant role in dental education. Given that dental students may encounter challenges, particularly in the cementation of all-ceramic restorations, a standardized protocol utilizing Erbium laser was implemented for non-invasive retrieval of miss-cemented restorations in dental student clinics. The retrospective analysis of the clinics with the same retrieval protocol would enhance the reliability of the data collection process and ensure consistency across cases. Lastly, a retrospective analysis is essential in establishing a baseline of clinical data before embarking on a prospective study. This baseline data provides valuable insights into the initial conditions and informs the design and objectives of the future prospective investigation.”

 2. The study should have standardization in inclusion criteria, like type of crown, location of crown, type of prosthesis like FDP, laminate or implant. All cannot be included in one study design. It can lead to many confounding variables and the impact is reduced.

RESPONSE: We appreciate this valuable insight. However, as a retrospective analysis, our study is designed to include all all-ceramic restorations/prostheses that have been removed by Erbium lasers. We recognize that the wide variety of prosthetic types, abutments, and cements may introduce increased confounding variables. Nevertheless, as the first and largest cohort of this kind, we aim to include all cases in this study to establish a comprehensive baseline for future prospective research that may adopt a more selective approach.

TEXT CHANGE: The following statement was added in the beginning of the Discussion.

“By encompassing the full spectrum of cases, our study seeks to provide a comprehensive overview and valuable insights into the clinical outcomes associated with Erbium laser removal of all-ceramic restorations/prostheses. The inclusion of diverse cases will contribute to a more robust understanding of the subject matter and pave the way for future investigations to target specific subgroups based on refined selection criteria.”

 3.Burning/ debonding with lasers are done, but how was the crown removed. It is done either done by conventional crown removers or with hand removal. The results denote for few crowns the time taken is more.It denotes the conventional removal was supported in removal. The whole study can become arguable.

RESPONSE: Thank you for providing this valuable insight. We have carefully incorporated the clarification regarding the removal protocol and timekeeping in the Methods section. Although this study is not prospective, we ensured strict adherence to the clinical protocol established in our previous in vitro studies (References #11-16). The prosthesis retrieval and timekeeping were consistently conducted following the established protocol. The additional clarification statements in the Methods section aim to further enhance the transparency and reliability of our research approach. We are grateful for your input, which has contributed to the refinement of our methodology and strengthened the integrity of our study.

TEXT CHANGE: The following statements were added in the Methods.

“After approximately 1-3 minutes of irradiation, we attempted to dislodge the prosthesis using a pair of hemostats with digital manipulation. If the prosthesis was not dislodged, the laser irradiation was continued. The removal protocol was standardized based on established methodologies from several previous studies [11-16].” 

“The prosthesis removal time was meticulously recorded as an integral part of the clinical protocol. The timing commenced from the initial application of laser irradiation and concluded upon the successful retrieval of the prosthesis, aligning with the methodology utilized in our previous in vitro studies [11-16].”

REVIEWER #2:

This clinical study evaluated the effectiveness of erbium laser in the removal of ceramic crowns. The research is valuable as it is an in vivo study, and the data has been presented well. This reviewer has a minor recommendation: The study included ceramic restorations with different light transmission properties. It would be beneficial to discuss the findings of a recent study that includes zirconia ceramic materials with different translucency properties: Birand, Cemil, and Sevcan Kurtulmus-Yilmaz. "Evaluation of Er, Cr: YSGG laser irradiation for debonding of zirconia hybrid abutment crowns from titanium bases." Lasers in Medical Science 37.6 (2022): 2675-2685.

RESPONSE: Thank you for your valuable feedback. We appreciate your recognition of the study's significance as an in vivo evaluation of erbium laser effectiveness in removing ceramic crowns. We have made efforts to present the data comprehensively.

We appreciate your recommendation regarding the inclusion of a recent study by Birand et al. (2022) that evaluates Er, Cr: YSGG laser irradiation for debonding zirconia hybrid abutment crowns from titanium bases. We incorporated a discussion as one of the limitation of the study.

TEXT CHANGE: The follow statement was added in the limitation of the study in the Discussion.

“Fourth, the presence of various types of ceramic compositions, materials, and levels of translucency could have influenced laser transmission, potentially impacting the retrievability and retrieval time of the restorations. [11-16,37]”

Reference #37 was added as suggested.

“37. Birand C, Kurtulmus-Yilmaz S. Evaluation of Er,Cr:YSGG laser irradiation for debonding of zirconia hybrid abutment crowns from titanium bases. Lasers Med Sci. 2022;37:2675-2685.”

---

## [Decision Letter · Decision Letter 1]

4 Dec 2023

Evaluation of the Effectiveness and Practicality of Erbium Lasers for Ceramic Restoration Removal: A Retrospective Clinical Analysis

PONE-D-23-15529R1

Dear Dr. Bencharit,

We’re pleased to inform you that your manuscript has been judged scientifically suitable for publication and will be formally accepted for publication once it meets all outstanding technical requirements.

Kind regards,

Mirza Rustum Baig

Academic Editor

PLOS ONE

Additional Editor Comments (optional):

Reviewers' comments:

Reviewer's Responses to Questions

**Comments to the Author**

1. If the authors have adequately addressed your comments raised in a previous round of review and you feel that this manuscript is now acceptable for publication, you may indicate that here to bypass the “Comments to the Author” section, enter your conflict of interest statement in the “Confidential to Editor” section, and submit your "Accept" recommendation.

Reviewer #1: All comments have been addressed

2. Is the manuscript technically sound, and do the data support the conclusions?

Reviewer #1: Partly

3. Has the statistical analysis been performed appropriately and rigorously? 

Reviewer #1: (No Response)

4. Have the authors made all data underlying the findings in their manuscript fully available?

Reviewer #1: Yes

5. Is the manuscript presented in an intelligible fashion and written in standard English?

Reviewer #1: Yes

6. Review Comments to the Author

Reviewer #1: Satisfied with the responses. The scientific impact of the manuscript has to be decided by the ediotorial board/ editor.

7. PLOS authors have the option to publish the peer review history of their article (what does this mean?). If published, this will include your full peer review and any attached files.

Reviewer #1: No

---

## [Editor Report · Acceptance letter]

6 Dec 2023

PONE-D-23-15529R1 

Evaluation of the Effectiveness and Practicality of Erbium Lasers for Ceramic Restoration Removal: A Retrospective Clinical Analysis 

Dear Dr. Bencharit:

I'm pleased to inform you that your manuscript has been deemed suitable for publication in PLOS ONE. Congratulations! Your manuscript is now with our production department. 

Kind regards, 

on behalf of

Dr. Mirza Rustum Baig 

Academic Editor

PLOS ONE